# Clinical and Laboratory Features of JAK2 V617F, CALR, and MPL Mutations in Malaysian Patients with Classical Myeloproliferative Neoplasm (MPN)

**DOI:** 10.3390/ijerph18147582

**Published:** 2021-07-16

**Authors:** Razan Hayati Zulkeflee, Zefarina Zulkafli, Muhammad Farid Johan, Azlan Husin, Md Asiful Islam, Rosline Hassan

**Affiliations:** 1Department of Haematology, School of Medical Sciences, Universiti Sains Malaysia, Kubang Kerian 16150, Malaysia; rhayatiz@usm.my (R.H.Z.); faridjohan@usm.my (M.F.J.); asiful@usm.my (M.A.I.); 2Hospital Universiti Sains Malaysia, Kubang Kerian 16150, Malaysia; azlanh@usm.my; 3Department of Internal Medicine, School of Medical Sciences, Universiti Sains Malaysia, Kubang Kerian 16150, Malaysia

**Keywords:** hematologic malignancies, chronic myeloproliferative neoplasm, molecular biology, polycythemia vera, essential thrombocythemia, primary myelofibrosis, JAK2V617F, calreticulin, MPL

## Abstract

Mutations of JAK2V617F, CALR, and MPL genes confirm the diagnosis of myeloproliferative neoplasm (MPN). This study aims to determine the genetic profile of JAK2V617F, CALR exon 9 Type 1 (52 bp deletion) and Type 2 (5 bp insertion), and MPL W515 L/K genes among Malaysian patients and correlate these mutations with clinical and hematologic parameters in MPN. Mutations of JAK2V617F, CALR, and MPL were analyzed in 159 Malaysian patients using allele-specific polymerase chain reaction, including 76 polycythemia vera (PV), 41 essential thrombocythemia (ET), and 42 primary myelofibrosis (PMF) mutations, and the demographics of the patients were retrieved. The result showed that 73.6% JAK2V617F, 5.66% CALR, and 27.7% were triple-negative mutations. No MPL W515L/K mutation was detected. In ET and PMF, the predominance type was the CALR Type 1 mutation. In JAK2V617F mutant patients, serum LDH was significantly higher in PMF compared to PV and ET. PV has a higher risk of evolving to post PV myelofibrosis compared to ET. A thrombotic event at initial diagnosis of 40.9% was high compared to global incidence. Only one PMF patient had a CALR mutation that transformed to acute myeloid leukemia. JAK2V617F and CALR mutations play an important role in diagnostics. Hence, every patient suspected of having a myeloproliferative neoplasm should be screened for these mutations.

## 1. Introduction

Myeloproliferative neoplasms (MPNs) are clonal hematopoietic disorders that harbor somatic mutations characterized by the excessive production of one or more terminally differentiated myeloid lineages. MPN is classified into (i) chronic myeloid leukemia (BCR-ABL1-positive), (ii) polycythemia vera (PV), (iii) essential thrombocythemia (ET), (iv) primary myelofibrosis (PMF), (v) chronic neutrophilic leukemia, (vi) chronic eosinophilic leukemia—not otherwise specified (NOS), and (vii) MPN—unclassifiable. Although most of the MPNs are indolent, MPNs have the potential to become thromboembolic and/or bleeding events and can transform into acute leukemia or myelofibrosis [1,2,3].

Mutations of *JAK2-V617F, CALR*, and *MPL* account for over 90% of MPN cases and are usually mutually exclusive to diagnosis. However, around 10% of those MPN driver mutations are still unknown, and these patients are defined as triple-negative MPN (TN-MPNs) [4]. The diagnostic driver gene mutation hallmark in MPN, especially in PV, is *JAK2-V617F* [5], while the second common mutation, mainly in ET and PMF, is the *CALR* mutation detected in 25% of MPN patients. The *MPL* mutation has been reported in 4–6% of ET and PMF individuals [6].

The Janus kinase 2 (*JAK2*) gene is mapped to chromosome 9p24.1 in humans [7]. In normal hematopoietic cells, *JAK2* is characterized by the presence of two homologous kinases: one is an enzymatically active kinase domain (JAK homology 1; JH1) and the other comprises a catalytically inactive pseudokinase domain (JH2) that inhibits the kinase activity of *JAK2* following ligand binding [8]. The most common mutation involves the substitution of valine with phenylalanine at position 617 in the JH2 domain (*JAK2-V617F*) located in exon 14, resulting in disrupts of normal inhibitory function towards the JH1 kinase domain. This mutation also induces excessive myeloid progenitor cell proliferation and activity by directly activating the JH1 domain through the SH2-JH2 link [9].

The calreticulin (*CALR*) chaperone protein-encoding gene is mapped on chromosome 19p13 in humans and contains nine exons. The *CALR* is a multifunctional protein that resides mainly in the endoplasmic reticulum and cytoplasm. It has several functions that contribute to calcium homeostasis, and it acts as a chaperone in interactions with *MPL* and JAK/STAT signaling, controlling cell proliferation and survival [10]. The most common mutations involved at either 52 base pair deletions (referred to as Type 1) or 5 base pair insertions (Type 2) are restricted to exon 9, which generates a +1 base-pair frameshift, eliminating most of the C-terminal domain and disrupting the Ca++ binding process [11].

The myeloproliferative leukemia (*MPL*) gene, mapped to chromosome 1p34, has 12 exons residing at the juxtamembrane region of the thrombopoietin receptor protein (TPOR). *MPL* is a regulator of thrombopoietin (TPO) levels; it is a negative feedback mechanism that eliminates TPO bound to the MPL receptor. *MPL* and TPO are two essential proteins for hematopoietic stem cell self-renewal and DNA repair [12]. The binding of TPO to the MPL receptor initiates signaling through JAK2. Several mutations of MPL have been identified. W515L (tryptophan-to-leucine substitution) and W515K (tryptophan-to-lysine substitution) are the most commonly involved mutations that occur within exon 10; they lead to the loss of auto-inhibition of the thrombopoietin receptor, which results in continuous activation.

Laboratory diagnosis of MPN is made based on peripheral blood count, bone marrow trephine biopsy morphology, biochemical investigations, BCR-ABL, and/or a positive finding for at least driver gene mutations or clonal proliferation (updated WHO 2016 criteria) [13,14]. Figure 1 shows the bone marrow trephine morphological features based on the updated WHO 2016 criteria for PV, ET, and PMF, respectively.

It is important to perform genetic testing to confirm the diagnosis of MPNs. To date, genetic testing is not done in many parts of the world to confirm MPNs. Therefore, this study aims to determine the genetic profile of *JAK2* V617F, *CALR* exon 9 Type 1 (52 bp deletion) and Type 2 (5 bp insertion), and *MPL* W515 L/K genes among Malaysian patients using allele-specific polymerase chain reaction and to correlate these mutations with clinical and laboratory parameters.

## 2. Materials and Methods

This was a cross-sectional and cohort study over five years. This study was approved by the Universiti Sains Malaysia Research Ethics Committee (USM/JEPeM 19100646) and carried out in accordance with the Declaration of Helsinki. In this study, 159 patient records were obtained from the Hematology Laboratory of Hospital Universiti Sains Malaysia (Hospital USM). The following inclusion selection criteria were applied: (1) 20 years old and above at the time of diagnosis and (2) diagnosis of MPN, as strictly defined by the criteria of the 2016 WHO classification guidelines. The diagnostic DNA samples were retrieved from the laboratory from 2016 until 2020. The clinical and laboratory data obtained were age, gender, white blood cell (WBC) and differential counts, hemoglobin (Hb), platelet (plt), hematocrit (HCT), lactate dehydrogenase (LDH), spleen and liver size, thrombotic and hemorrhagic event, progression of disease, and *JAK2V617F* mutation status. Mutational analyses (*CALR* and *MPL*) were performed using archived extracted genomic DNA and newly diagnosed samples. Isolation of genomic DNA from newly collected peripheral blood was carried out using QIAamp DNA Blood Mini Kits (QIA Symphony SP, Qiagen Company, Hilden, Germany), adhering to the protocol described by the manufacturer [15]. *JAK2V617F, CALR*, and *MPL* mutational status were determined via allele-specific oligonucleotide primers polymerase chain reaction (PCR) with to the 5′ and 3′ ends of exon 14 of *JAK2*, exon 9 of *CALR*, and exon 10 of *MPL*, respectively, which were done through the NCBI nucleotide BLAST program (https://blast.ncbi.nlm.nih.gov/Blast.cgi; accessed on 8 July 2019). The following accession numbers of DNA, primers, PCR reaction, and PCR reaction volume were used for PCR amplification of the *JAK2V617F, CALR*, and *MPL* mutations (Appendix A). The PCR reaction mixture was up in the total volumes of 25 μL from constituents, and the reactions were placed in a Veriti PCR thermal cycler following the PCR profile from the reference stated. The PCR products were electrophoresed on 1.5% agarose gel for 30 min (Figure 2). Both *CALR* and *MPL* gene mutations were confirmed by Sanger sequencing [16].

The results were analyzed by Statistical Package for the Social Sciences (SPSS) Statistics for Windows (Version 25.0; IBM Corp., Armonk, NY, USA). For parametric data, comparisons between patient characteristics were performed by the chi-square method for categorical variables and the independent *t*-test for the numerical data. Additionally, the Mann–Whitney U-test and Fisher’s exact test were used for nonparametric and qualitative variables, respectively. *p*-values less than 0.05 were considered to indicate statistically significant data.

## 3. Results

The clinical and laboratory features of patients with chronic MPN are summarized in Table 1. A total of 159 patients were studied; 89 (56%) were male and 70 (44%) were female, with mean and standard deviation ages of 57.18 (13.52) years. Of the 159 patients, MPN was classified into 76 (47.8%) PV, 41 (25.8%) ET, and 42 (26.4%) PMF. Mean hemoglobin, leukocyte and differential counts, platelet counts, and LDH are shown in Table 1. One PMF patient transformed to acute myeloid leukemia after more than 10 years from diagnosis, and 5 (3.1%) PV patients progressed to myelofibrosis.

Of the 76 patients diagnosed with PV, 49 (64.5%) were male and 27 (35.5%) were female, with a mean age of 58.66 (13.90) years (range, 20–80 years). Splenomegaly was present in 36.8% of patients. The second type of MPN was primary myelofibrosis, with 42 patients recruited. The mean age was 57.66 (11.49) years (range, 32 to 79). Splenomegaly was present in 61.9% of patients. Forty-one patients were diagnosed with ET; 46% were male and 54% were female. The mean age was 53.98 (14.43) years (range, 23–86 years). Splenomegaly was present in 36.6% of patients. Thrombosis at the time of diagnosis was a frequent manifestation, with an overall prevalence of 40.9%.

Table 2 shows the prevalence of three driver gene mutations in different types of MPN. A total of 117 (73.6%) MPN patients were detected with the *JAK2V617F* mutation. The detection rates of the *JAK2V617F* mutation in PV, ET, and PMF patients were 86.8% (66/76), 70.7% (29/41), and 52.3% (22/42), respectively. Nine cases (5.66%) of *CALR* gene exon 9 mutations were detected, which were found to be mutually exclusive in ET and PMF patients. Based on the MPN subtypes, *CALR* gene mutations were detected in 7.31% of ET patients, and all were Type 1. In contrast, in PMF, the *CALR* gene mutation was found in 14.3% of patients; 66.7% and 33.3% were Type 1 and Type 2, respectively. No *MPL* W515L/K mutations were detected in all types of MPN. Additionally, 27.7% of MPN patients were triple-negative. No coexistence of two or more gene mutations was detected.

Table 3 demonstrates the breakdown of MPN subtypes according to *JAK2V617F* mutation status. The group of PV *JAK2V617F-*positive patients presented at an older age (*p* < 0.05), with significantly high platelet counts and high leukocyte counts (*p* < 0.001); 24.2% and 37.9% of patients with PV with the *JAK2V617F* mutation had hepatomegaly and splenomegaly, respectively. ET patients with the *JAK2V617F* mutation had higher hemoglobin levels (*p* < 0.005) and increased leukocyte and granulocyte counts. No patient progressed to the blast phase or was transformed to the fibrosis stage.

*JAK2V617F* mutation-positive patients were also analyzed within different subtypes of MPN patients (Table 3). It is as expected that hemoglobin and platelet counts were significantly higher in PV (*p* < 0.05) and ET (*p* = 0.001), respectively, compared with other subtypes. Serum LDH was significantly higher in PMF (*p* = 0.008) compared to PV and ET.

Table 4 summarizes the clinical and laboratory features of ET and PMF patients, stratified according to the *JAK2V617F, CALR* mutation and triple-negative status. Univariate analysis was performed on patients with ET. It was noted that those with *CALR* mutations were associated with higher platelet counts, higher LDH levels, and lower thrombotic events. For PMF, the univariate analysis in patients with the *CALR* mutation showed very high serum LDH and increased platelet counts, but they were protected from thrombosis. None had any thrombotic events to date. One patient with *CALR* mutations had leukemic transformation to acute myeloid leukemia.

## 4. Discussion

The *JAK2V617F* mutation plays a decisive role as a screening tool for the diagnosis of MPN, particularly for PV. To date, additional mutations such as *CALR* and *MPL* mutations are also useful to confirm (Philadelphia) Ph-MPN. Despite the stringent diagnostic criteria outlined by the WHO edition, the diagnosis of *JAK2V617F*-negative in PV, ET, and PMF is still challenging to clinicians because 27.7% of our MPN patients were triple-negative. Thus, the morphological examination of bone marrow remains the fundamental investigation to diagnose Ph-MPN, especially triple-negative MPN.

There is a slight gender bias in MPNs, with a predominance of males over females, in agreement with other literature. However, there is some inconsistency between the subtypes [17]. In this study, the mean age for *JAK2V617F*-mutated patients is older compared to those with the wild type. This probably reflects that the mutation was acquired throughout the aging process.

Hitherto, there have been limited data on *JAK2V617F, CALR*, and *MPL* mutations and their correlation with clinical and hematological parameters in the region. The subtypes of MPN of 47.8%, 25.8%, and 26.4% for PV, ET, and PMF, respectively, detected in this study, were similar to prospective cohort studies among Asian and German populations [18,19].

In our study, *JAK2V617F* was the most prevalent mutation in MPN, accounting for 73.6% of our population. The detection rates of the *JAK2V617F* mutation in PV, ET, and PMF patients were 86.8%, 70.7%, and 52.3%, respectively. These findings were similar to those in Chinese and Sudanese populations, where the *JAK2V617F* mutation was 85–91% in PV, 54% in ET, and 65.8% in PMF [20,21].

In this study, 5.66% of *CALR* mutations were detected among MPN patients who were diagnosed as PMF and ET without *JAK2V617F* and *MPL* mutations. None of our PV patients had *CALR* mutations. These findings were consistent with a Slovenian cohort study by Belcic et al., which showed that about 4.4% of their patients had *CALR* mutations [11]. In contrast to Korean, Chinese, and American patients, PMF and ET patients showed higher percentages of *CALR* mutations, estimated to range from 12.6% to 89%, compared to our study range of 7.3–14.3% [20,22,23]. This was possibly due to differences in (1) the analytical sensitivities of the methods used and (2) the geographical distribution of the study population. Similar to other studies, the predominant type of *CALR* was Type 1 (77.78%) [22].

Interestingly, our results showed that none of our MPN patients had *MPL W515L/K* mutations. Lieu and Eldeweny et al. also reported that *MPL W515L/K* could not be found in 88 Taiwanese and 60 Egyptian patients with MPN [24,25]. Only 1–4% of *MPL* mutations were reported in two studies conducted among Korean and Chinese populations [20], [22]. Likewise, the *MPL* mutation was detected in 4–6% of ET and PMF in Iranian and Turkish populations [6,26]. This small discrepancy may be due to differences in ethnicity. None of the double mutations for *MPL* and *CALR* have been acknowledged in our study, similar to the finding of Lang et al. [27].

The frequency of our triple-negative mutations was 27.7%, which was similar to a Korean study (20%) [22]. Otherwise, the frequencies of the triple-negative mutations were found to be lower (10–15%) in Indian and Slovenia studies [11,28]. Other types of clonal mutations, such as *ASLX1, EZH2, TET2, IDH1/IDH2, SRSF2*, and *SF3B1* genes, may be demonstrated in PMF patients [29]. Apart from that, the method for mutational detection in this study was an allele-specific test that was designed to assess only hotspot mutations [30]. On the other hand, ET patients may have small mutant allele fractions that are lower than the limit of the detection method, leading to false-negative results.

There was a lack of clinically relevant patients with splenomegaly. However, a more extensive cohort study among the Busan Korean population found that larger spleen volumes at presentation may predict poor prognosis and a risk to fibrotic transformation or leukemia transformation [31,32]; 36.8% of our PV patients had splenomegaly in line with previous literature, and 6.57% had transformed to myelofibrosis.

The hematological parameters of PV patients in this study showed typical panmyelosis, as described in the WHO revised edition of 2016. The *JAK2* mutation is a key player in autonomous red cell production and the stimulation of myeloid and megakaryocytic lineages. The substitution of valine to phenylalanine at the 617 position of the *JAK2* gene leads to the continuous activation of cytokine receptors, i.e., erythropoietin, thrombopoietin, and granulocyte colony-stimulating factors, resulting in an amplified production of RBCs, leukocytes, and platelets [33].

Our ET and PMF patients, similar to a previous study of Spanish, Korean, and Mexican patients, demonstrated that those with CALR mutations had higher platelet counts than those with other mutations [22,34,35].

The majority of PMF *JAK2V617F*-mutated patients in this study showed monocytosis. It is noted that monocytosis is associated with short survival and rapid disease progression [36,37,38]. However, the relationship between monocytosis and survival analysis in this study is unknown due to the lack of survival data.

In this study, PMF patients had very high serum LDH compared to *JAK2V617F-*negative ET and PV patients. Serum LDH was currently reconstituted as part of minor diagnostic criteria for PMF [13,38]. Nonetheless, increased LDH is not specific to PMF, as it can also be seen in ET and PV [39]. The utility of serum LDH is not only limited to the diagnosis of MPN, but it can also be used as one of the prognostic markers. A study by Shah and Busque et al. proposed that marked elevation of serum LDH  ≥ 1000 U/L may predict a shorter survival outcome. The basis of this proposal lies in the fact that LDH is known to be associated with rapid cell turnover and ineffective hematopoiesis and hemolytic processes occurring in the spleen [40,41].

MPN is known to have an increased risk of thrombosis. We observed that thrombotic events at initial diagnosis in 40.9% of our MPN patients were higher than the global incidence of 20% [42]. The risk of thrombosis was higher among ET, followed by PV, in *JAK2V617F-*positive patients. The pathophysiology behind the thrombotic risk in PV is due to increased blood viscosity, secondary to high hematocrit levels, and the migration of platelets to the vessel wall and subsequent platelet activation [43]. As in ET, it may be due to increased myeloid production, activation of leukocytes, and endothelial dysfunction [44]. It is postulated that MPN clonal cells induce the release of inflammatory cytokines, which results in procoagulant states. This finding is supported by blood-clotting markers, i.e., thrombin antithrombin complex, thrombomodulin, von Willebrand factor/factor V111, and prothrombin fragment 1 + 2 [45]. The majority of our MPN patients had thrombotic events that were commonly due to arterial thrombosis compared to venous thrombosis; this is consistent with most studies [42].

Interestingly, this study showed that PV has a higher risk of evolving into post PV myelofibrosis compared to ET at the median of 54 (24–132) months. Only one PMF patient with the *CALR* mutation transformed to acute myeloid leukemia after 20 years of follow-up. This finding can be due to de novo or secondary to underlying disease (MPN); further investigation on the mechanism of leukemogenesis is needed. More cases are required to confirm the progression of the disease. None of the ET patients had disease progression or leukemic transformation in this study. Therefore, distinguishing subtypes play a major role in risk stratification.

## 5. Conclusions

The practical purposes of the current study are summarized as follows. (1) Identifying these molecular genes is important in the diagnosis of Ph- MPN, especially since the presentation may overlap with the reactive causes and other myelodysplastic/myeloproliferative diseases. (2) *CALR* is the second-most frequently detected mutation in MPN; therefore, it is part of the diagnostic tools for MPN, especially in *JAK2V617F-*negative PMF and ET. (3) Bone marrow morphology remains the backbone of MPN diagnosis because up to 20% of triple negatives have an absence of MPN-definite molecular markers.

Despite a small cohort, the results of this study should be heeded as this is the first report on the frequency of *JAK2V617F, CALR*, and *MPL* mutations in relation to the biological and clinical features of MPN among Malaysians.

## Figures and Tables

**Figure 1 ijerph-18-07582-f001:**
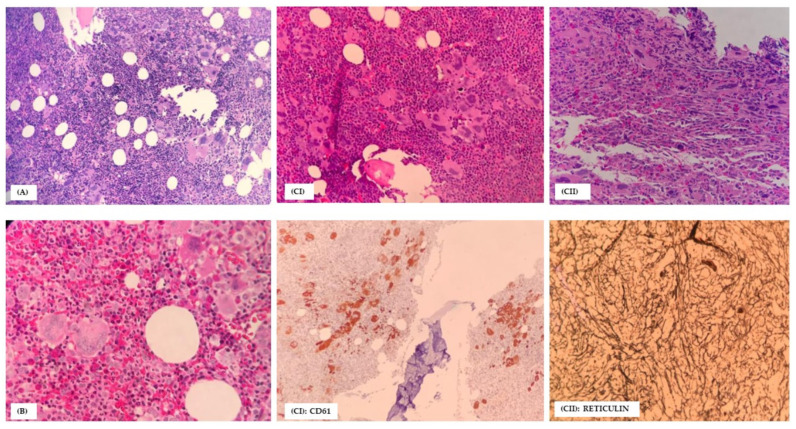
H& E (**A**) Polycythemia vera; hypercellularity with panmyelosis(especially erythroid precursors and prominent megakaryocytes). The florid megakaryocytes admixture of small, normal to large pleomorphic morphology and arranged in loose clusters. Dilated sinus containing red blood cells. (**B**) Essential thrombocythemia; normal in cellularity, proliferation of abnormally giant or large form of atypical megakaryocytes, abundant cytoplasm, scattered throughout the marrow spaces. Erythropoiesis and granulopoiesis were within normal. (**C**) Primary myelofibrosis; (**CI**) pre-PMF: hypercellularity with a proliferation of atypical megakaryocytic arranged in dense or large clusters and usually adjacent to vascular sinus and bony trabeculae. Megakaryocytes morphology (high nuclear-cytoplasmic ratio, abnormal chromatin clumping, (**CII**) Overt PMF: Variable cellularity with (sometimes) absence of hematopoietic cells, dense reticulin, atypical megakaryocytes, usually in sheets and within the dilated sinuses.

**Figure 2 ijerph-18-07582-f002:**
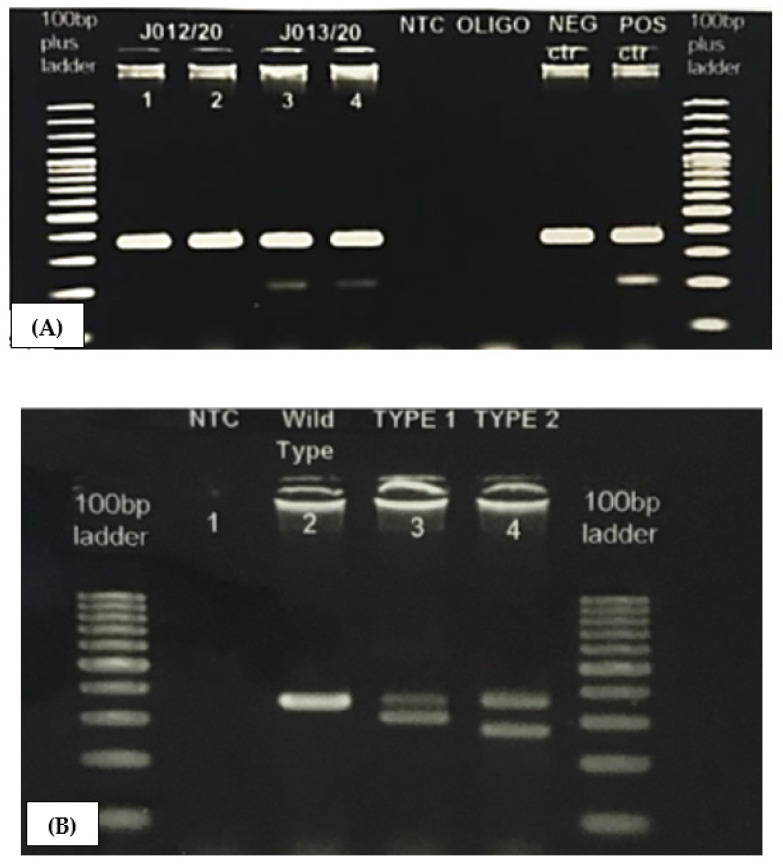
PCR products with gel electrophoresis. Electrophoresis of amplified PCR products to detect (**A**) *JAK2 V617F* mutations: Lanes 1–2 are normal cases with only wild-type at 364 bp, Lanes 3–4 are positive for the *JAK2 V617F* mutation, with wild-type at 364 bp and a mutant band at 203 bp; (**B**) *CALR* mutations: Lane 2 is a normal case with only wild-type at 357 bp, Lane 3 is positive for *CALR* Type 1 (302 bp), and Lane 4 is positive for CALR Type 2 (272 bp).

**Table 1 ijerph-18-07582-t001:** Clinical and laboratory features of patients (*n* = 159) with chronic MPN.

		Disease Subtype		
Characteristic	MPN (*n* = 159)	PV (*n* = 76)	ET (*n* = 41)	PMF (*n* = 42)
Sex (M/F)	89/70	49/27	19/22	21/21
		**Mean (SD)**		
Age at diagnosis	57.18 (13.52)	58.66 (13.90)	53.98 (14.43)	57.66 (11.49)
Hb (g/dL) (Range)	14.85 (3.79)	17.26 (2.97) (7.7–23.4)	13.54 (2.1) (9–18)	11.68 (3.12) (6.5–20.2)
Hct (%)	46.95 (12.55)	54.78 (10.14)	41.74 (7.35)	37.65 (10.67)
WBC (×10^9^/L) (Range)	17.15 (14.02)	17.22 (10) (4.02–55.57)	15.02 (14.58) (7.18–97.7)	17.16 (16.77) (2.7–76.4)
Platelets (×10^9^/L) (Range)	645.82 (399.76)	599.36 (375.4) (94–1752)	901.29 (289.29) (305–1597)	534.39 (446.3) (22–2035)
Monocyte (×10^9^/L)	1.02 (1.44)	0.78 (0.32)	0.65 (0.34)	1.25 (1.51)
Basophil (×10^9^/L)	0.23 (0.55)	0.24 (0.64)	0.10 (0.08)	0.31 (0.55)
Eosinophil (×10^9^/L)	0.61 (1.69)	0.50 (0.52)	0.34 (0.26)	0.58 (1.54)
LDH (U/L) ULN > 400	867.45 (827.08)	704.63 (406.24)	624.07 (336.75)	1141.7 (1331.01)
Hepatomegaly *	44 (27.7)	17 (22.4)	7 (17.0)	20 (47.6)
Splenomegaly *	69 (43.4)	28 (36.8)	15 (36.6)	26 (61.9)
Thrombosis *	65 (40.9)	32 (42.1)	15 (36.6)	18 (42.9)
Hemorrhage *	2 (1.26)	1 (1.31)	1 (2.43)	0
Transformed *	6 (3.77)	5 (6.57)		1 (2.38)

* All of these variables are presented as *n* (%).

**Table 2 ijerph-18-07582-t002:** Prevalence of *JAK2^V617F^*, *CALR*, and *MPL* gene mutations.

Characteristic	ET [*n* = 41 (%)]	PMF [*n* = 42 (%)]	PV [*n* = 76 (%)]
*JAK2V617F*	29 (70.7)	22 (52.3)	66 (86.8)
*CALR* exon 9+	3 (7.3)	6 (14.3)	0
*MPL* exon 10+	0	0	0
Triple negative	9 (21.9)	14 (33.4)	-

**Table 3 ijerph-18-07582-t003:** Clinicohematological parameters of *JAK2^V617F^* positive and negative cases according to MPN subtypes.

Parameter	PV Mean (SD)/*n* (%)			ET Mean (SD)/*n* (%)			PMF Mean (SD)/*n* (%)					
	*JAK2V617F* Positive *n* = 66 (a)	*JAK2V617F* Negative *n* = 10	*p-Value*	*JAK2V617F* Positive *n* = 29 (b)	*JAK2V617F* Negative *n* = 12	*p-Value*	*JAK2V617F* Positive *n* = 22 (c)	*JAK2V617F* Negative *n* = 20	*p-Value*	*p-Value (a vs. b)*	*p-Value (b vs. c)*	*p-Value (a vs. c)*
† Gender, Male *	40 (60.6)	9 (90)	0.087	12 (41.3)	7 (58.3)	0.322	8 (36.4)	13 (65)	0.064	0.117	0.778	0.778
‡ Age, years	60.36 (13.49)	47.40 (11.62)	0.005	54.79 (12.77)	52 (18.32)	0.579	60.05 (10.59)	55.45 (12.09)	0.20	0.988	0.346	0.346
‡ Hb (g/dL)	17.2 (3.15)	17.7 (1.25)	0.368	14.1 (2.0)	12.2 (1.75)	0.007	11.73 (3.7)	11.54 (2.4)	0.847	0.028	0.003	0.003
‡ HCT (%)	55.06 (10.78)	52.93 (3.62)	0.231	43.53 (7.37)	37.43 (5.42)	0.14	36.64 (10.79)	38.37 (10.74)	0.607	0.033	0.054	0.054
‡ WBC (×10^9^/L)	18.5 (10.01)	8.82 (3.76)	<0.001	16.71 (17.1)	10.93 (1.75)	0.253	20.86 (16.04)	16.61 (17.69)	0.42	0.02	0.672	0.672
‡ Platelets (×10^9^/L)	653.56 (373.15)	241.6 (70.51)	<0.001	875.31 (290.3)	964.1 (289.34)	0.378	698.64 (467.57)	426.2 (383.76)	0.47	0.001	0.092	0.092
‡ Monocyte (×10^9^/L)	0.78 (0.34)	0.76 (0.17)	0.790	0.69 (0.37)	0.57 (0.27)	0.383	1.37 (1.13)	1.42 (1.89)	0.927	0.279	0.047	0.002
‡ Basophil (×109/L)	0.27 (0.70)	0.05 (0.02)	0.347	0.11 (0.08)	0.07 (0.05)	0.154	0.41 (0.62)	0.2 (0.44)	0.256	0.07	0.054	0.448
‡ Eosinophil (×109/L)	0.54 (0.56)	0.28 (0.22)	0.172	0.35 (0.21)	0.34 (0.36)	0.974	1.01 (2.03)	0.17 (0.17)	0.087	0.398	0.217	0.158
‡ LDH (U/L) ULN > 400	763.81 (370.73)	670 (551.52)	0.516	602.6 (267.3)	66.8 (450.2)	0.659	1162.37 (1087.03)	1350.19 (1605.37)	0.684	0.058	0.008	0.008
† Hepatomegaly *	16 (24.2)	1 (10)	0.441	3 (10.3)	4 (33.3)	0.165	13 (59.1)	7 (35)	0.118	0.119	0.002	<0.001
† Splenomegaly *	25 (37.9)	3 (30)	0.737	10 (34.5)	5 (41.7)	0.73	17 (77.2)	9 (45)	0.31	0.752	0.484	0.002
† Thrombotic events, * - Artery - Vein	26 (39.4) 21 (80.8) 5 (19.2)	6 (60) 6 (100) 0	0.306	12 (41.3) 11 (91.7) 1 (8.3)	3 (25) 1 (33.3) 2 (66.7)	0.48	7 (31.8) 5 (71.4) 2 (28.6)	11 (55) 1 (9.1) 10 (90.9)	0.129	0.856	<0.001	0.484
† Transformed *	5 (7.5)	0	1	0	0	-	0	1 (Type 1)	0.488	-	-	-

(i) * All of these variables are presented with *n* (%). (ii) All the results were within 95% CI. ‡ Independent ***t***-test. † Pearson chi-square test. ***p* < 0.05** is statistically significant.

**Table 4 ijerph-18-07582-t004:** Clinical and laboratory features of 41 Malaysian patients with PMF and 42 patients with ET, stratified according to mutation profiles.

	PMF; *n* = 42						ET; *n* = 41					
Variables *n*(%) Median (Range)	*JAK2V617F* Mutation *n* = 22 (a)	CALR Mutation *n* = 6 (b)	Triple Negative *n* = 14 (c)	*p-Value* *(a vs. b)*	*p-Value* *(b vs. c)*	*p-Value* *(a vs. c)*	*JAK2V617F* Mutation *n* = 29 (a ^1^)	CALR Mutation *n* = 3 (b ^1^)	Triple Negative *n* = 9 (c ^1^)	*p-Value* *(a * ^1^ * vs. b * ^1^ *)*	*p-Value* *(b * ^1^ * vs. c * ^1^ *)*	*p-Value* *(a * ^1^ * vs. c * ^1^ *)*
^1^ Gender, Male *	8 (36.4)	2 (33.3)	11 (78.6)	1.0	0.122	0.013	12 (41.4)	2 (66.7)	5 (55.6)	0.568	1.0	0.703
^2^ Age, years	60 (43–79)	61 (38–75)	55 (32–73)	0.779	0.457	0.177	51 (33–86)	69 (41–81)	52 (23–75)	0.348	0.165	0.336
^2^ Hemoglobin, g/dL	11.25 (6.5–20.20)	11.25 (7.1–12.4)	11.55 (7.4–16.5)	0.695	0.264	0.537	14.4 (9–18)	13.2 (9.1–14.4)	11.8 (9.90–14.30)	0.184	0.782	0.004
^2^ Hematocrit, %	36.1 (18.8–60)	36.05 (33–77. 0)	36.55 (23.8 46.5)	0.433	0.934	0.795	44.4 (18–54.6)	39.3 (27.6–42.8)	39.8 (30.1–43.5)	0.065	0.782	0.003
^2^ WBC, ×10^9^/L	17.03 (3.6–76.4)	14.69 (8.38–72.28)	10.25 (2.65–61.09)	0.654	0.161	0.035	11.7 (7.18–97.77)	12.04 (9–13.89)	10.43 (7.70–12.96)	0.923	0.518	0.400
^2^ Platelets, *n* × 10^9^/L	639.5 (84–2035)	723.5 (208–1532)	254 (33–541)	0.737	0.048	0.002	851 (305–1597)	1051 (749–1129)	1014 (472–1454)	0.539	1.0	0.420
^2^ Monocyte ×10^9^/L	0.98 (0.29–3.99)	1.39 (0.72–8.33)	0.78 (0.28–1.22)	0.126	0.012	0.196	0.58 (0.3–1.44)	0.42 (0.32–0.61)	0.68 (0.15–0.92)	0.244	0.302	0.834
^2^ LDH, U/L ULN > 400	852 (397–4145)	2071 (614–5842)	530 (262–1355)	0.056	0.009	0.060	534.5 (351–1459)	616 (450–662)	449.0 (354–1880)	0.546	0.424	0.809
^1^ Splenomegaly *	17 (77)	4 (66.7)	5 (35.7)	0.622	0.336	0.013	10 (34.5)	2 (66.7)	3 (33.3)	0.54	0.523	1.0
^1^ Liver *	13 (59.1)	2 (33.3)	5 (35.7)	0.372	1.0	0.171	3 (10.3)	1 (33.3)	3 (33.3)	0.34	1.0	0.131
^1^ Thrombotic events *	7 (31.8)	0	11 (78.6)	0.288	0.001	0.006	12 (41.4)	1 (33.3)	3 (33.3)	1.0	-	-
^1^ Transformation *	0	1 (Type 1)	0				0	0	0			

(i) * All of these variables are presented with *n* (%); (ii) all the results were within 95% CI. ^1^ Fisher’s exact test; ^2^ Mann–Whitney U-test; ***p* < 0.05** is statistically significant.

## Data Availability

The data presented in this study are available on request from the corresponding author.

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
