# Peer review of "Clinical and Laboratory Features of JAK2 V617F, CALR, and MPL Mutations in Malaysian Patients with Classical Myeloproliferative Neoplasm (MPN)"

_ijerph, 2021, doi:10.3390/ijerph18147582_

Round 1

Reviewer 1 Report

This paper, which investigated gene mutations in Malaysian MPN patients, may be the first in Malaysia. However, the number of patients studied is too small to determine the exact mutation frequency. In addition, all the data shown are the same as in the published papers and are not scientifically novel. This paper is suitable for publication in academic journals in Malaysia, but not for publication in international journals.

Major
1. The authors apply the concept of triple-negative not only to ET and PMF but also to PV, but there is no concept of triple-negative PV.

2. In the abstract, the authors compare LDH levels in JAK2 mutation-positive PMF patients with LDH levels in ET and PV patients without JAK2 mutations, but the significance of this comparison is unclear.

3. In the abstract, the authors state that the JAK2V617F and CALR mutations play important roles in diagnosis and prognosis prediction, but no data showing the significance of the mutations in prognosis are included in this paper.

Reviewer 2 Report

Zulkeflee et al. present a paper looking at JAK2, CALR and MPL mutations in a cohort of 159 Malaysian MPN patients. They present mutation frequency and clinical features of these patients, also broken down into the different sub-types of MPN.

Overall, the paper is scientifically sound and the data are well presented. The paper is suitable for publication, subject to the specific points below being addressed. Some English language editing is also required, but the paper is generally easy to understand.

Major points:

It is unclear in the paper which CALR and MPL mutations you were looking for. I gathered from your primer table that you were looking for CALR Type I and Type II, and MPL W515L and W515K mutations. It would be good to state this earlier in the paper, and include some information about what these mutations do. (This was done well for JAK2 V617F.)

Minor points:

Line 16: I would simplify “allele-specific 16 oligonucleotide-polymerase chain” in what is meant to be a “simple summary”.

Line 52: It would be good to clarify here whether the CALR and MPL mutations are specific point mutations (like JAK2 V617F), or a variety of mutations within those genes. (See also the major point above.)

Line 113: Do you mean 1.5% agarose gel?

Line 154: “No MPL mutations” presumably means the two that you tested for. Is it possible that there could be others? Are any other mutations known from the literature? If there are, you should at least note that you did not test for these for clarity.

Line 163: If this result is not statistically significant, then it should not be reported.

Line 56-64 (second block of numbering): It’s not clear how the two parts of this paragraph relate.

Line 94 (second block of numbering): Please explain why you see the discrepancy between this study and other studies.

Reviewer 3 Report

Authors reported a cohort of 159 Malaysian patients affected by myeloproliferative neoplasm (MPN) and investigated the occurrence of JAK2V617F, CALR, and MPL gene mutations correlating genetics findings with clinical and hematologic characteristics.

The incidence of triple negative MPN is high as compared to literature data, this could indicate two possible explanations: 1) molecular methods used for identifying gene mutations were not satisfying 2) WHO criteria employed for diagnosis were not correctly applied.

The study is to be appreciated as it underlines the importance of genetic tests worldwide for NPM diagnosis, however the molecular methods used for the analysis are not fully satisfactory. In details:

  1. Nowadays, the analysis of JAK2 V617F mutations is mainly carried out by molecular quantitative methods (real-time PCR or digital PCR) given the important prognostic significance attributed to the mutation allelic burden;
  2. The method used for revealing CALR type 1 mutation cannot be properly defined ASO-PCR as it is based on the use of primers that bind wild type regions upstream and downstream of the 52 bp deletion;
  3. the electrophoretic pattern related to CALR type 2 mutation is not convincing (figure 2B, lane 9, ctr type 2) as in addition to the wild type allele, a smaller fragment is observed, similar to the type 1 mutation and not a larger fragment as expected following an insertion (type 2 mutation). This anomalous pattern could explain why CALR type 2 mutation is observed at a lower frequency in the analysed patients. The authors should provide a compelling explanation for the assay used to identify CALR type 2 mutation.

Round 2

Reviewer 1 Report

1) For line 167 on page 2 and line 41 on page 10, if the concept of triple-negative is not applied to PV, the frequency is not 20.8%.

2) For the p-values in Tables 3 and 4, the authors should clearly state the test method from which the p-values are derived.

Reviewer 3 Report

All points have been sufficiently addressed

Author Response

Thank you for your reviews and comments.